# Testing gene-environment interactions for rare and/or common variants in sequencing association studies

**Zihan Zhao[1], Jianjun Zhang[2], Qiuying Sha[3], Han Hao[2]***

**1** Texas Academy of Mathematics & Science, University of North Texas, Denton, TX, United States of America, **2** Department of Mathematics, University of North Texas, Denton, TX, United States of America, **3** Department of Mathematical Sciences, Michigan Technological University, Houghton, MI, United States of America

* Han.Hao@unt.edu

## Abstract

The risk of many complex diseases is determined by a complex interplay of genetic and environmental factors. Advanced next generation sequencing technology makes identification of gene-environment (GE) interactions for both common and rare variants possible. However, most existing methods focus on testing the main effects of common and/or rare genetic variants. There are limited methods developed to test the effects of GE interactions for rare variants only or rare and common variants simultaneously. In this study, we develop novel approaches to test the effects of GE interactions of rare and/or common risk, and/or protective variants in sequencing association studies. We propose two approaches: 1) testing the effects of an optimally weighted combination of GE interactions for rare variants (TOW-GE); 2) testing the effects of a weighted combination of GE interactions for both rare and common variants (variable weight TOW-GE, VW-TOW-GE). Extensive simulation studies based on the Genetic Analysis Workshop 17 data show that the type I error rates of the proposed methods are well controlled. Compared to the existing interaction sequence kernel association test (ISKAT), TOW-GE is more powerful when there are GE interactions' effects for rare risk and/or protective variants; VW-TOW-GE is more powerful when there are GE interactions' effects for both rare and common risk and protective variants. Both TOW-GE and VW-TOW-GE are robust to the directions of effects of causal GE interactions. We demonstrate the applications of TOW-GE and VW-TOW-GE using an imputed data from the COPDGene Study.

## Introduction

The etiology of many diseases is characterized by the interplay between genetic and environment factors. For example, anthracyclines are one of the most effective classes of chemotherapeutic agents currently available for cancer treatment. The therapeutic potential of

**Data Availability Statement:** The data of the COPDGene study is third party data. Others can access the data by submitting applications to dbGap for dbGaP Study Accession: (phs000179/

HMB and phs000179/DS-CSRD). The authors confirm that others would be able to access the data in the same manner as the authors and that they did not have any special access privileges that others would not have.

**Funding:** Research reported in this publication was supported by the National Human Genome Research Institute of the National Institutes of Health under Award Number R15HG008209 to QS. The content is solely the responsibility of the authors and does not necessarily represent the official views of the National Institutes of Health. The Genetic Analysis workshops are supported by NIH grant R01 GM031575 from the National Institute of General Medical Sciences. Preparation of the Genetic Analysis Workshop 17 Simulated Exome Data Set was supported in part by NIH R01 MH059490 and used sequencing data from the 1000 Genomes Project (www.1000genomes.org). This research used data generated by the COPDGene study, which was supported by NIH grants U01HL089856 and U01HL089897. The COPDGene project is also supported by the COPD Foundation through contributions made by an Industry Advisory Board comprised of Pfizer, AstraZeneca, Boehringer Ingelheim, Novartis, and Sunovion. The funders had no role in study design, data collection and analysis, decision to publish, or preparation of the manuscript.

**Competing interests:** The authors have declared that no competing interests exist. Q Sha was supported by the National Human Genome Research Institute of the National Institutes of Health under Award Number R15HG008209. The content is solely the responsibility of the authors and does not necessarily represent the official views of the National Institutes of Health. This does not alter our adherence to PLOS ONE policies on sharing data and materials. The Genetic Analysis workshops are supported by NIH grant R01 GM031575 from the National institute of General Medical Sciences. Preparation of the Genetic Analysis Workshop 17 Simulated Exome Data Set was supported in part by NIH R01 MH059490 and used sequencing data from the 1000 Genomes Project (www.1000genomes.org). This research used data generated by the COPDGene study, which was supported by NIH grants U01HL089856 and U01HL089897. The COPDGene project is also supported by the COPD Foundation through contributions made by an Industry Advisory Board comprised of Pfizer, AstraZeneca, Boehringer Ingelheim, Novartis, and Sunovion. The funders had no role in study design, data collection and analysis, decision to publish, or preparation of the

anthracyclines, however, is limited because of their strong dose-dependent relation with progressive and irreversible cardiomyopathy leading to congestive heart failure. Both gene hyaluronan synthase 3 (*HAS*3) and gene *CUGBP* Elav-like family member 4 (*CELF*4) modify the risk of anthracycline on the development of anthracycline-related cardiomyopathy [1, 2]. A genome-wide gene environment interaction analysis indicates that gene *EBF*1 plays together with stress associated with cardiovascular disease. Additionally, gene *EBF*1 not only shows gene-by-stress interaction effect for hip circumference but also indicates gene-by-stress interaction effects for waist circumference, body mass index (BMI), fasting glucose, type II diabetes, and common carotid intimal medial thickness (CCIMT) [3].

To date, most of the successful findings in gene-environment (GE) interactions are for common genetic variants. There has been very limited success in findings for rare variants' GE interactions. This is often attributed to study design issues, such as sample size or population heterogeneity [4]. Lack of statistical methodology on rare variants' GE also contributes to the limitations.

Rare variants, which are usually defined as genetic variants with minor allele frequency (MAF) less than 5% (or 1%), may play an important role in studying the etiology of complex human diseases. Numerous statistical methods have been developed for testing the main effects of rare variants, such as the sequence kernel association test (SKAT) [5], the combined multivariate and collapsing (CMC) method [6], the weighted sum statistic (WSS) [7], and Testing the effect of an Optimally Weighted combination of variants (TOW) [8].

To our knowledge, limited methods have been developed for testing GE interactions in sequencing association studies. Existing methods for assessing common variants by environment interactions, such as the gene-environment interactions association test (GESAT) [9] are less powerful when naively applied to rare variants [10]. To test rare variants by environment interactions, [10] developed the interaction sequence kernel association test (ISKAT) to assess the effects of rare variants by environment interactions. As ISKAT considers the special weights Beta(MAF;1, 25), the beta distribution density function with parameters 1 and 25 evaluated at the sample MAF, which is the recommended weight for ISKAT when there is no prior information, ISKAT may lose power when the MAFs of causal variants are not in the range (0.01,0.035) [11].

In this article, to test for rare and/or common variants and environment interactions in sequencing association studies, we develop two novel methods: 1) Testing the Optimally weighted combination of GE interactions for rare variants (TOW-GE); 2) testing effects of weighted combination of GE interactions for both rare and common variants (variable weight TOW-GE, refer to this statistic as VW-TOW-GE). Both TOW-GE and VW-TOW-GE are robust to directions of effects of causal GE interactions. We evaluate the performance of the proposed methods via simulation studies and real data analysis using the imputed sequencing data from the COPDGene Study.

## Methods

Consider $n$ unrelated individuals sequenced in a testing region with $m$ genetic variants. In the testing region, we are interested in testing the effects of $p$ rare variants ($p < m$) by environment interactions on a trait, which can be a quantitative or a qualitative trait. For ease of presentation, we only consider a single environmental factor. The method can be easily extended to the case when there are multiple environmental factors. For individuals $i = 1, \ldots, n$, let $y_i$ denote the trait, $X_i = (x_{i1}, \ldots, x_{iq})^T$ denote the $q$ covariates, $G_i = (g_{i1}, \ldots, g_{ip})^T$ denote genotypes for the

manuscript. This does not alter our adherence to
PLOS ONE policies on sharing data and materials.

$p$ rare variants in a genomic region (a gene or a pathway) and $E_i$ as the environmental factor. Let $\boldsymbol{S}_i = (E_i g_{i1}, \ldots, E_i g_{ip})^T$ be a vector of variants by environment interaction terms for the $i^{th}$ individual.

We use the generalized linear model (GLM) to model the relationship between the trait values $y_i$ and covariates $\boldsymbol{X}_i$, genotypes $\boldsymbol{G}_i$, environmental factor $E_i$ and GE interactions, $\boldsymbol{S}_i$:

$$
\begin{aligned}
g(\mathbb{E}(y_i|\boldsymbol{X}_i, \boldsymbol{G}_i, E_i)) & = \boldsymbol{X}_i^T \boldsymbol{\alpha}_1 + E_i \alpha_2 + \boldsymbol{G}_i^T \boldsymbol{\alpha}_3 + \boldsymbol{S}_i^T \boldsymbol{\beta} \\
& = \tilde{\boldsymbol{X}}_i^T \boldsymbol{\alpha} + \boldsymbol{S}_i^T \boldsymbol{\beta}
\end{aligned}
\tag{1}
$$

where $g(\cdot)$ is a canonical link function. Two commonly used models under the generalized linear model framework are the linear model with the identity link for a continuous or quantitative trait, and the logistic regression model with the Logit link for a binary trait. $\boldsymbol{\alpha}_1$, $\alpha_2$, $\boldsymbol{\alpha}_3$, and $\boldsymbol{\beta}$ are defined as $q \times 1$ coefficient vector of covariate, the coefficient of environmental factor, $p \times 1$ coefficient vector of genotype and $p \times 1$ coefficient vector of GE interactions for the $i^{th}$ individual and the trait, respectively. Let $\tilde{\boldsymbol{X}}_i = (\boldsymbol{X}_i^T, E_i, \boldsymbol{G}_i^T)^T$ and $\boldsymbol{\alpha} = (\boldsymbol{\alpha}_1, \alpha_2, \boldsymbol{\alpha}_3)^T$. Testing the association between the trait and the rare variants by environment interactions is equivalent to testing the null hypothesis $H_0: \boldsymbol{\beta} = 0$.

We develop a score test by treating $\boldsymbol{\alpha}$ as nuisance parameters and then adjust both the trait value $y_i$ and $\boldsymbol{S}_i$ for the covariates $\boldsymbol{X}_i$, the genotypic score $\boldsymbol{G}_i$, and the environmental variable $E_i$ by applying linear regression. Denote $\tilde{y}_i$ as the residual of $y_i$ and $\tilde{\boldsymbol{S}}_i = (\tilde{s}_{i1}, \ldots, \tilde{s}_{ip})$ as the residual of $\boldsymbol{S}_i$, regressed on $\tilde{\boldsymbol{X}}_i$. Then, the relationship between $\tilde{y}_i$ and $\tilde{\boldsymbol{S}}_i$ can be modeled by the GLM:

$$
g(\mathbb{E}(\tilde{y}_i|\tilde{\boldsymbol{S}}_i)) = \beta_0^* + \tilde{\boldsymbol{S}}_i^T \boldsymbol{\beta}^*
\tag{2}
$$

Testing $H_0: \boldsymbol{\beta} = 0$ in (1) is equivalent to testing $H_0: \boldsymbol{\beta}^* = 0$ in (2) (Sha et al., [8]). Here, we utilize a weight selection scheme proposed by Sha et al. [8] on our new model to test the effect of a weighted combination of GE, $\tilde{s}_i = \sum_{j=1}^p w_j \tilde{s}_{ij}$. Following Sha et al. [12], we propose the following score test statistic under the generalized linear model:

$$
\begin{aligned}
S(w_1, \ldots, w_p) & = n \frac{\left(\sum_{i=1}^n (\tilde{y}_i - \bar{\tilde{y}})(\tilde{s}_i - \bar{\tilde{s}})\right)^2}{\sum_{i=1}^n (\tilde{y}_i - \bar{\tilde{y}})^2 \sum_{i=1}^n (\tilde{s}_i - \bar{\tilde{s}})^2} \\
& = n \frac{\left(\sum_{j=1}^p w_j \sum_{i=1}^n (\tilde{y}_i - \bar{\tilde{y}})(\tilde{s}_{ij} - \bar{\tilde{s}}_j)\right)^2}{\sum_{i=1}^n (\tilde{y}_i - \bar{\tilde{y}})^2 \sum_{i=1}^n (\tilde{s}_i - \bar{\tilde{s}})^2}
\end{aligned}
$$

Because GE interactions for rare variants are essentially independent, we have:

$$
\begin{aligned}
\sum_{i=1}^n (\tilde{s}_i - \bar{\tilde{s}})^2 & = \sum_{j=1}^p \sum_{l=1}^p w_j w_l \sum_{i=1}^n (\tilde{s}_{ij} - \bar{\tilde{s}}_j)(\tilde{s}_{il} - \bar{\tilde{s}}_l) \\
& \approx \sum_{j=1}^p w_j^2 \sum_{i=1}^n (\tilde{s}_{ij} - \bar{\tilde{s}}_j)^2
\end{aligned}
$$

Thus, as a function of $(w_1, \ldots, w_p)$, the score test statistic $S(w_1, \ldots, w_p)$ reaches its maximum $S_0(w_1^0, \ldots, w_p^0) = n \sum_{i=1}^n (\tilde{y}_i - \bar{\tilde{y}})(\tilde{s}_i^0 - \bar{\tilde{s}}^0) / \sum_{i=1}^n (\tilde{y}_i - \bar{\tilde{y}})^2$ when $w_j^0 = \sum_{i=1}^n (\tilde{y}_i - \bar{\tilde{y}}) \times (\tilde{s}_{ij} - \bar{\tilde{s}}_j) / \sum_{i=1}^n (\tilde{s}_{ij} - \bar{\tilde{s}}_j)^2$ and $\tilde{s}_i^0 = \sum_{j=1}^p w_j^0 \tilde{s}_{ij}$.

Similarly, we define the statistic to Test the effect of the Optimally Weighted combination of GE interactions (TOW-GE), $\sum_{j=1}^{p} w_j^0 \tilde{s}_{ij}$, as:

$$T_{TOW-GE} = \sum_{i=1}^{n} (\tilde{y}_i - \bar{\tilde{y}})(\tilde{s}_i^0 - \bar{\tilde{s}}^0) \tag{3}$$

which is equivalent to $S_0(w_1^0, \ldots, w_p^0)$, where $\sum_{i=1}^{n} (\tilde{y}_i - \bar{\tilde{y}})^2$ can be viewed as a constant when we use a permutation test to evaluate p-values.

The optimal weight $w_j^0$ is equivalent to $w_j^{0*} = \rho(\tilde{y}, \tilde{s}_j)/\sqrt{\sum_{i=1}^{n} (\tilde{s}_{ij} - \bar{\tilde{s}}_j)^2}$, where $\rho(\tilde{y}, \tilde{s}_j)$ is the correlation coefficient between $\tilde{y} = (\tilde{y}_1, \ldots, \tilde{y}_n)$ and $\tilde{s}_j = (\tilde{s}_{1j}, \ldots, \tilde{s}_{nj})$. From the expression of $w_j^{0*}$, we can see that it is proportional to $\rho(\tilde{y}, \tilde{s}_j)$ and thus $w_j^0$ will put large weights to the GE interactions that have strong associations with the trait and also adjust for the direction of the association. Simultaneously, $w_j^{0*}$ is proportional to $1/\sqrt{\sum_{i=1}^{n} (\tilde{s}_{ij} - \bar{\tilde{s}}_j)^2}$ and $w_j^0$ will put large weights to GE interactions with small variations which are common in GE interactions for rare variants.

TOW-GE focuses primarily on rare variants by environment interactions and it may lose power because of the small weights on common variants by environment interactions. Thus, to test the GE interactions' effects of both rare and common variants, we propose the following variable weight TOW-GE denoted as VW-TOW-GE. We first divide GE interactions into two parts based on rare or common variants and then we apply TOW-GE to the two parts separately. Let $T_\lambda = \lambda \frac{T_r}{\sqrt{var(T_r)}} + (1 - \lambda) \frac{T_c}{\sqrt{var(T_c)}}$ where $T_r$ and $T_c$ denote the test statistics of TOW-GE for GE interactions' effects of rare and common variants, respectively. $\lambda$ is a tuning parameter. Denote $p_\lambda$ as the p-value of $T_\lambda$, and then the test statistic of VW-TOW-GE is defined as $T_{VW-TOW-GE} = \min_{0 \leq \lambda \leq 1} p_\lambda$. In this study, we use a simple grid search method to choose the tuning parameter $\lambda$ and minimize the p-value. Divide the interval [0, 1] into $K$ subintervals of equal-length. Let $\lambda_k = k/K$ for $k = 0, 1, \ldots, K$. Then, $\min_{0 \leq \lambda \leq 1} p_\lambda = \min_{0 \leq k \leq K} p_{\lambda_k}$.

The p-value of $T_{VW-TOW-GE}$ can be evaluated by permutation tests following similar permutation tests for variable weight TOW (VW-TOW) proposed by [8]. Suppose that we perform B times of permutations. In each permutation, we randomly shuffle the trait values. Let $T_r^{(b)}$ and $T_c^{(b)}$ denote the values of $T_r$ and $T_c$, respectively, based on the $b^{th}$ permuted data, where $b = 0$ represents the original data. Based on $T_r^{(b)}$ and $T_c^{(b)}$ ($b = 0, 1, 2, \ldots, B$), we can calculate $T_{\lambda_k}^{(b)}$ for $b = 0, 1, 2, \ldots, B$ and $k = 0, 1, 2, \ldots, K$, where $var(T_r)$ and $var(T_c)$ are estimated using $T_r^{(b)}$ and $T_c^{(b)}$ ($b = 0, 1, 2, \ldots, B$). Then, we transfer $T_{\lambda_k}^{(b)}$ to $p_{\lambda_k}^{(b)}$ by

$$p_{\lambda_k}^{(b)} = \frac{\#\{T_{\lambda_k}^{(d)} : T_{\lambda_k}^{(d)} > T_{\lambda_k}^{(b)} \, for \, d = 0, 1, 2, ..., B\}}{B}$$

Let $p^{(b)} = \min_{0 \leq k \leq K} p_{\lambda_k}^{(b)}$. Then, the p-value of $T_{VW-TOW-GE}$ is given by

$$\frac{\#\{p^{(b)} : \, p^{(b)} < p^{(0)} \, for \, b = 0, 1, 2, ..., B\}}{B}$$

## Simulation

We compared the performance of our proposed methods with the interaction sequence kernel association test (ISKAT) [10], the modified WSS for testing the effects of GE interactions [7] and the modified CMC method for testing the effects of GE interactions [6]. In this study, the

rank sum test used by WSS and the $T^2$ test used by CMC were replaced with the score test based on residuals $\tilde{y}_i$ and $\tilde{s}_{ij}$. The empirical Mini-Exome genotype data provided by the Genetic Analysis Workshop 17 (GAW17) is used for simulation studies. The dataset contains genotypes of 697 unrelated individuals on 3,205 genes. Because gene *ELAVL4* in GAW17 was used to simulate GE interaction's effect on quantitative trait $Q_1$ which follows a normal distribution, we chose gene *ELAVL4* in our simulation study. Gene *ELAVL4* has 10 variants, containing 8 rare variants and 2 common variants. Rare variants in the simulation are defined with MAF $< 0.05$.

To evaluate type I error, we generate trait values independent of GE interactions (e.g. $\boldsymbol{\beta}_1 = 0$ and $\beta^c = 0$) by using the model:

$$Y_1 = 0.5X_1 + 0.5X_2 + E\alpha_1 + \boldsymbol{G}^T\boldsymbol{\alpha}_2 + \boldsymbol{S}^T\boldsymbol{\beta}_1 + S_c\beta_1^c + \epsilon_1$$

where $\epsilon_1$ follows a normal distribution with mean as 0 and variances as $\sigma_1^2 = 1$; $\alpha_1 = 0.015$; $\boldsymbol{S}$ is GE interactions for rare variants and $S_c$ is GE interaction for a common variant. We consider two covariates: a standard normal covariate $X_1$ and a binary covariate $X_2$ with $P(X_2 = 1) = 0.5$. The environmental factor E is assumed to be continuous following a standard normal distribution.

For type I error evaluation, we consider two different cases: 1) testing the effects of GE interactions for rare variants; 2) testing the effects of GE interactions for both rare and common variants. For each case, we consider two scenarios: (a) with main effect; (b) without main effect in the model. When the main effects exist, we set the magnitudes of vector $\boldsymbol{\alpha}_2$ as 0.3 and the sign of each coefficient is random sampled from $(-1, 1)$. When main effects do not exist, we set $\boldsymbol{\alpha}_2 = 0$.

For power comparisons, the phenotype is generated using similar settings to type I evaluation except for existing GE interactions' effects. We compare the power of TOW-GE, ISKAT, WSS and CMC to test rare variant GE interactions' effects considering two scenarios: (a) including main effects, $\boldsymbol{\alpha}_2 \neq 0$ for rare variants; (b) no main effects, $\boldsymbol{\alpha}_2 = 0$ for rare variants. We vary the number of non-zero in the vector $\boldsymbol{\beta}_i$, the proportion of non-zero in $\boldsymbol{\beta}_i$ that are positive, and the magnitudes of the non-zero $\beta_{ij}$. We set the magnitudes of the non-zero $\beta_{ij}$'s as $|\beta_{ij}| = c$, and increase c from 0.1 to 0.5. In each simulation scenario, p-values are estimated by 10,000 permutations and 1,000 replicated samples.

## Simulation results

The empirical type I error rates are shown in Tables 1 and 2. For 10,000 replicated samples, the 95% confidence intervals for type I error rates of nominal levels as 0.05, 0.01 and 0.001 are

**Table 1. Type 1 error rates for testing the effects of GE interactions of rare variants in the presence of main effects (top panel) and in the absence of main effects (bottom panel) (n = 2000).**

| | $\boldsymbol{\alpha}$-level | TOW-GE | ISKAT | WSS | CMC |
|---|---|---|---|---|---|
| **With main effect** | | | | | |
| $n = 2000$ | 0.05 | 0.042 | 0.066 | 0.047 | 0.027 |
| | 0.01 | 0.01 | 0.017 | 0.011 | 0.005 |
| | 0.001 | 0.001 | 0.009 | 0.000 | 0.000 |
| **Without main effect** | | | | | |
| $n = 2000$ | 0.05 | 0.054 | 0.066 | 0.050 | 0.043 |
| | 0.01 | 0.006 | 0.014 | 0.012 | 0.012 |
| | 0.001 | 0.000 | 0.004 | 0.000 | 0.000 |

**Table 2. Type 1 error rates for testing the effects of GE interactions for both rare and common variants in the presence of main effects (top panel) and in the absence of main effects (bottom panel) (n = 2000).**

| | $\alpha$-level | TOW-GE | ISKAT | WSS | CMC | VW-TOW-GE |
|---|---|---|---|---|---|---|
| **With main effect** | | | | | | |
| n = 2000 | 0.05 | 0.053 | 0.062 | 0.055 | 0.040 | 0.052 |
| | 0.01 | 0.007 | 0.013 | 0.017 | 0.009 | 0.011 |
| | 0.001 | 0.002 | 0.002 | 0.001 | 0.000 | 0.002 |
| **Without main effect** | | | | | | |
| n = 2000 | 0.05 | 0.051 | 0.056 | 0.048 | 0.049 | 0.058 |
| | 0.01 | 0.006 | 0.012 | 0.012 | 0.011 | 0.014 |
| | 0.001 | 0.001 | 0.003 | 0.001 | 0.001 | 0.002 |

(0.046, 0.054), (0.008, 0.012) and (0.0004, 0.0016), respectively. When there are (a) main effects, e.g. $\alpha_2 \neq 0$, TOW-GE, VW-TOW-GE, ISKAT and WSS control type I error rates well and the burden test CMC tends to have very conservative type I error rates (top panel of Tables 1 and 2). When there are (b) no main effects. e.g. $\alpha_2 = 0$, all methods can control type I error rates well (bottom panel of Tables 1 and 2).

The results for testing the effects of GE interactions of rare variants when including main effect and no main effect are given in Figs 1 and 2, respectively. In both of these two scenarios, we consider the sample size as 2000 without a GE interaction of a common variant. We do not apply VW-TOW-GE here because it is designed for existing GE interactions' effects of both common and rare variants. The top, middle, and bottom panels in Figs 1 and 2 provide results for three cases, e.g. when there are 2, 6 and 8 non-zero $\beta_{ij}$'s, respectively. The left and right panels of Figs 1 and 2 present for two cases, e.g. 50% of the $\beta_{ij}$ are positive and 100% of the $\beta_{ij}$ are positive, respectively. For each plot, we vary c, the magnitudes of the non-zero $\beta_{ij}$. As shown in the four plots for the case when 50% of the $\beta_{ij}$ are positive, TOW-GE is more powerful than the other three tests. For the case when 100% of the $\beta_{ij}$ are positive, WSS is relatively more powerful than TOW-GE since all the GxEs have the same direction of effects. TOW-GE is more powerful than the other two tests. However, WSS is very sensitive to the directions of effects due to aggregation of GE interactions directly. Among the four tests (TOW-GE, ISKAT, WSS and CMC) in the two different cases, CMC is the least powerful test. CMC loses power as it gives GE interactions of common variants large weights, and thus GE interactions of common neutral variants will introduce large noise.

Power comparisons of the five tests (TOW-GE, VW-TOW-GE, ISKAT, WSS and CMC) for testing GE interaction effects for both rare and common variants are given in Fig 3. For each scenario in Fig 3, we vary c from 0.02 to 0.1 and set 50% of the $\beta_{ij}$ as positive. Simultaneously, we set the coefficient of a common variant by environment interaction $\beta_i^c$ as positive and the magnitudes of $\beta_i^c$ as twice of $\beta_{ij}$ which is the coefficient of a rare variant by environment interaction. From Fig 3, we can see that VW-TOW-GE is the most powerful test. CMC is the second most powerful test as CMC puts large weights on GE interactions of common variants and gains power increment when the GE interaction of a common variant plays an important role as the causal effect. WSS is the least powerful test, which loses power because it puts very small weight on the GE interaction of the common variant.

TOW-GE, VW-TOW-GE, and ISKAT can all be considered as quadratic statistics which have reasonable power across a wide range of alternative hypothesis. The three methods are robust to the different directions of the GE interaction effects. We perform a further assessment for the three methods. Fig 4 shows the results. When there are causal effects of GE interactions for both common and rare variants, VW-TOW-GE outperforms TOW-GE and

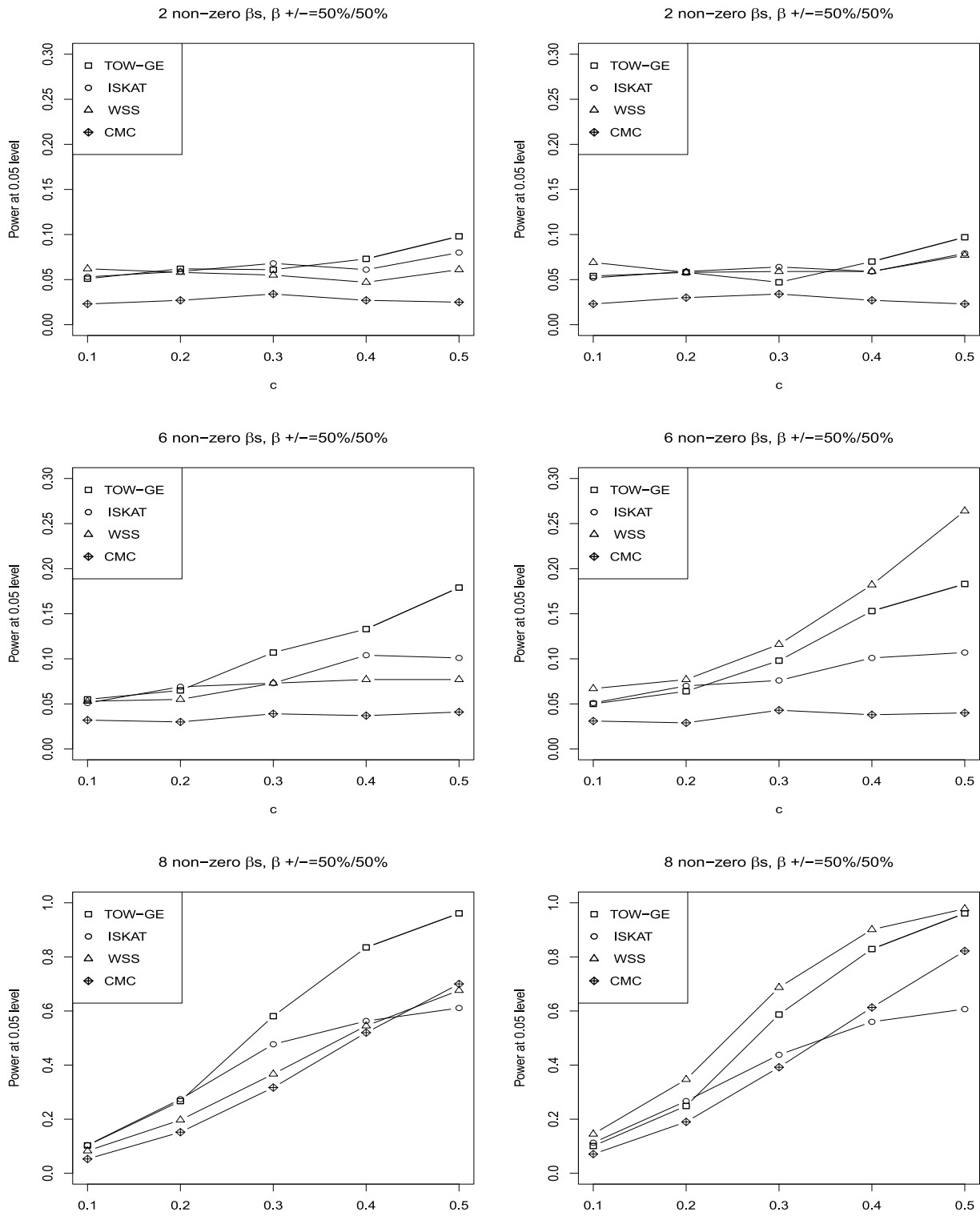

**Fig 1. Power comparisons of the four tests (TOW-GE, ISKAT, WSS and CMC) for testing GE interaction effects for rare variants on a continuous outcome when there are main effects (n = 2000 and the significance level of $\alpha = 0.05$).**

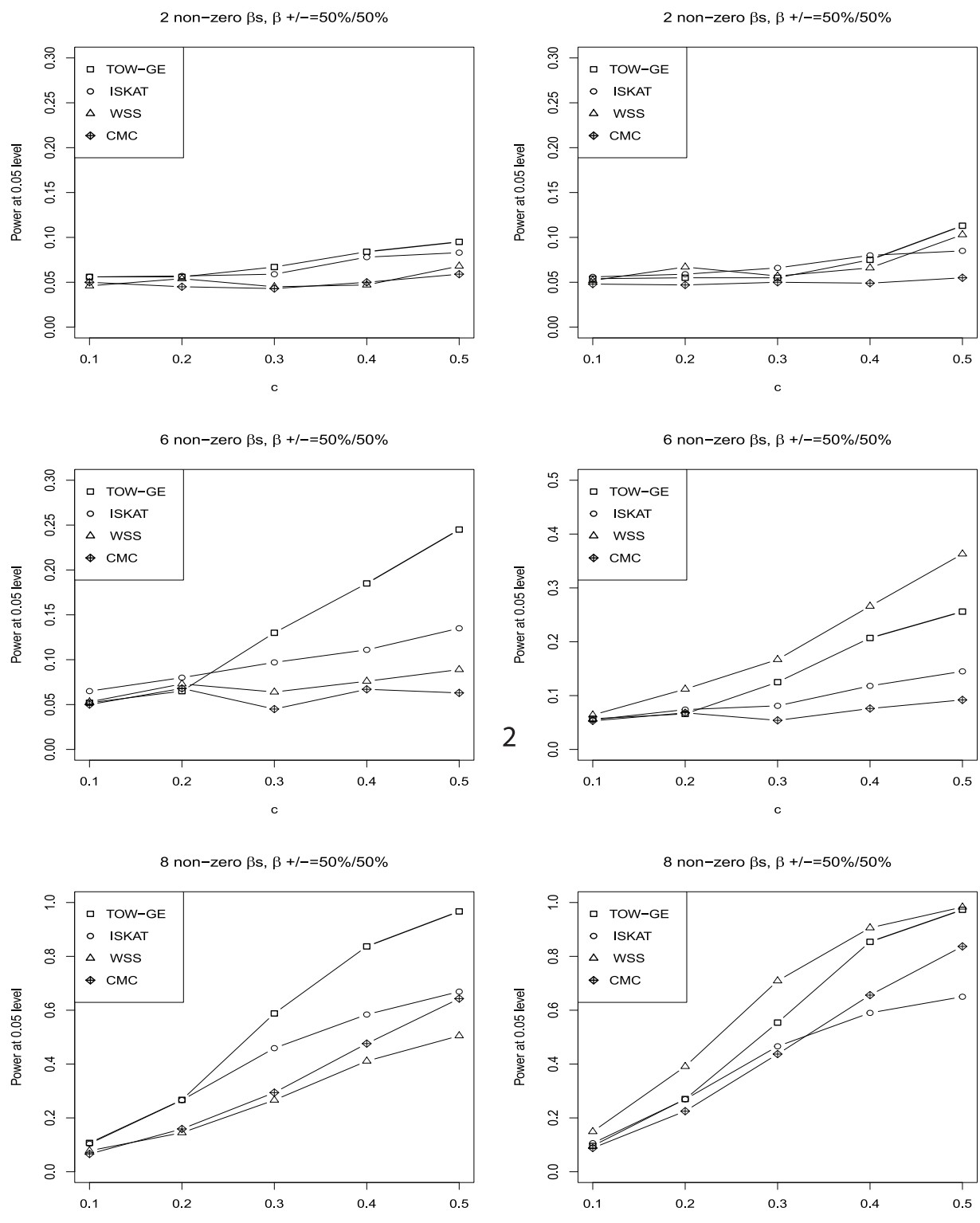

**Fig 2. Power comparisons of the four tests (TOW-GE, ISKAT, WSS and CMC) for testing GE interaction effects of rare variants on a continuous outcome when there are no main effects (n = 2000, significance level of $\alpha$ = 0.05).**

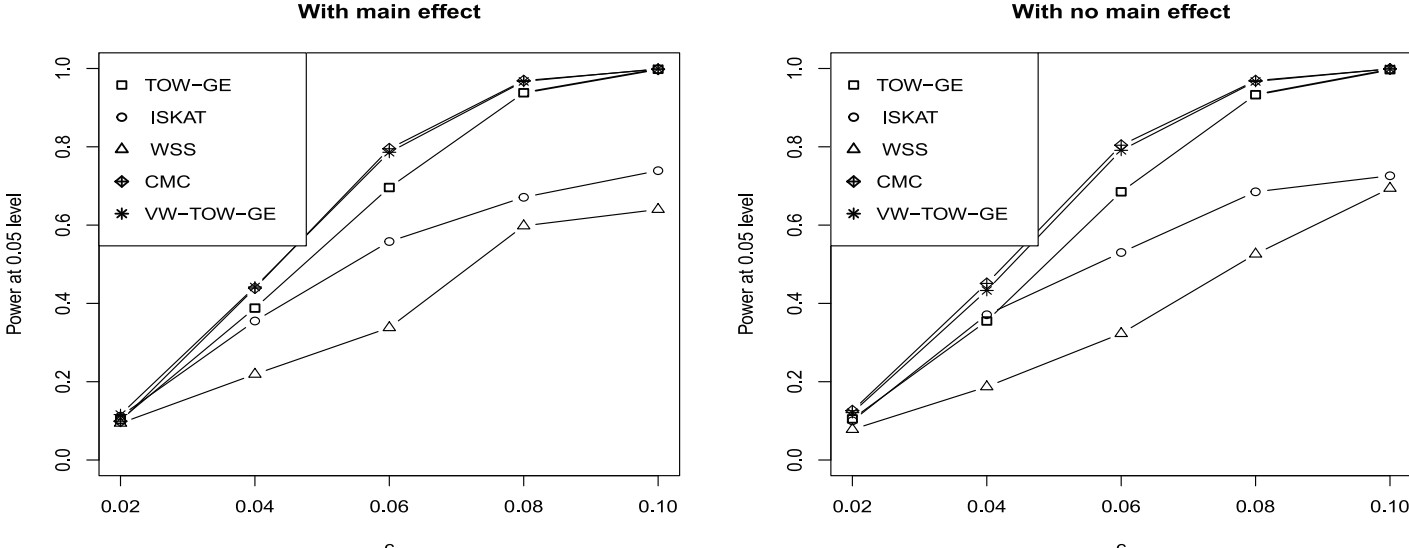

**Fig 3. Power comparisons of the five tests (TOW-GE, ISKAT, WSS, CMC and VW-TOW-GE) for testing GE interaction effects for both rare and common variants on a continuous outcome (n = 2000 and the significance level of $\alpha$ = 0.05).** Left panel: With main effect; Right panel: With no main effect.

ISKAT. TOW-GE is more powerful than ISKAT except when the magnitude of the GE interactions is less than 0.04.

## Real data analysis

Chronic obstructive pulmonary disease (COPD) is one of the most common lung diseases characterized by long term poor airflow and is a major public health problem [13]. It is a complex disease which is influenced by genetic factors, environmental influences, and genotype-environment interactions. We have known that cigarette smoking is the major environmental

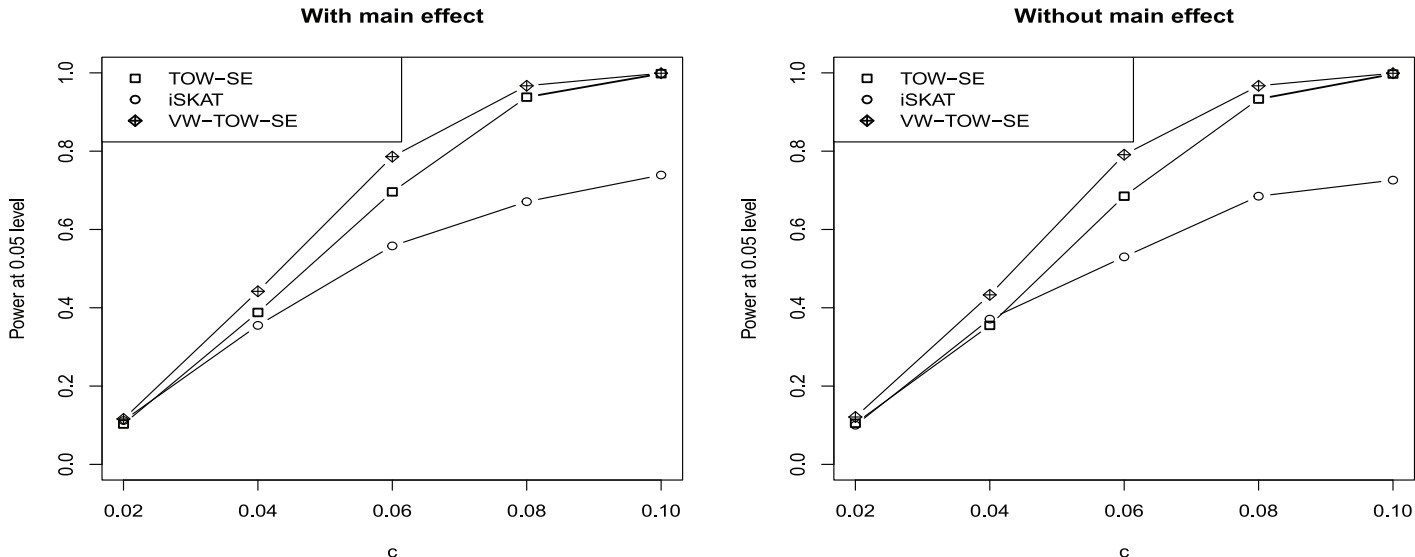

**Fig 4. Power comparisons of the three quadratic tests (TOW-SE, iSKAT, and VW-TOW-SE) for testing GE interaction effects of both rare and common variants on a continuous outcome (n = 2000, the significance of $\alpha$ = 0.05).** Left panel: With main effect; Right panel: Without main effect.

determinant of COPD [14]. Several genes have been suggested to play a role in the presence of a gene-by-smoking interaction term. Specifically, [15] reported that the 30-repeat allele of *HMOX*1 was associated with COPD in presence of a gene-by-smoking (pack-years) interaction term. [14] presented that the *GSTM*1 gene was associated with severe chronic bronchitis in heavy smokers and an association of the *TNF*—308A allele with COPD was found in a Taiwanese population. [15] reported that the *SFTPB* Thr131Ile polymorphism was associated with COPD, but only in the presence of a gene with an environment interaction. The SNP rs2292566 in gene *EPHX*1 was associated with COPD only in presence of a gene-by-smoking (pack-years) interaction. [16] showed that two SNPs in the promoter region of *TGFB*1 (rs2241712 and rs1800469) and one SNP in exon 1 of *TGFB*1 (rs1982073) were significantly associated with COPD among smokers in a COPD case control study.

The COPDGene Study is a multi-center genetic and epidemiologic investigation to study COPD [17]. Participants in the COPDGene Study gave consent for the use of data collected during the study in downstream analyses. This study is sufficiently large and appropriately designed for analysis of COPD. In this study, we consider more than 5,000 non-Hispanic Whites (NHW) participants where the participants have completed a detailed protocol, including questionnaires, pre- and post-bronchodilator spirometry, high-resolution CT scanning of the chest, exercise capacity (assessed by six-minute walk distance), and blood samples for genotyping. The participants were genotyped using the Illumina OmniExpress platform. The genotype data have gone through standard quality-control procedures for genome-wide association analysis detailed at http://www.copdgene.org/sites/default/files/GWAS_QC_Methodology_20121115.pdf. We imputed the COPD genotype data using the EUR haplotypes from the 1000 Genome Project as references.

Based on the literature of COPD [18, 19], we selected 7 key quantitative COPD-related phenotypes, including FEV1 (% predicted FEV1), Emphysema (Emph), Emphysema Distribution (EmphDist), Gas Trapping (GasTrap), Airway Wall Area (Pi10), Exacerbation frequency (ExacerFreq), Six-minute walk distance (6MWD), and one qualitative phenotypes (case-control disease status denoted as COPD in following tables). 3 covariates, including BMI, Age and Sex and one environmental factor (Pack-Years) were considered in our analysis. EmphDist is the ratio of emphysema at -950 HU in the upper 1/3 of lung fields compared to the lower 1/3 of lung fields where we did a log transformation on EmphDist in the following analysis, referred to [18]. In the analysis, participants with missing data in any of these phenotypes were excluded.

To evaluate the performance of our proposed method on a real data set, we applied all of the 5 methods (TOW-GE, ISKAT, WSS, CMC, and VW-TOW-GE) to six COPD associated genes (*HMOX*1, *GSTM*1, *TGFB*1, *TNF*, *SFTPB*, and *EPHX*1) through an interaction with cigarette smoking. In the analysis, we removed the extreme rare SNPs (MAF<0.001) in any genotypic variants and missing value in any of the 7 phenotypes and 3 covariates. We considered three different scenarios: (1) main effect; (2) gene-by-smoking interaction with main effect and (3) gene-by-smoking interaction without main effect. When we considered only the main effect, we used five existing methods (TOW-GE, SKAT, WSS, CMC, and VW-TOW) which are specifically designed for testing the main effect of a gene. We adopted $10^4$ permutations for our methods and used 0.05 as the significance level.

The results for testing association between COPD and gene *HMOX*1 and *GSTM*1 are summarized in Tables 3 and 4 respectively. The results for testing association between COPD and gene *TGFB*1, *TNF*, *SFTPB*, and *EPHX*1 are summarized in S1–S4 Tables. At gene *HMOX*1, both TOW-GE and modified WSS verified significant GE intecation effects without main effect for two traits Emph and Pi10. ISKAT and VW-TOW-GE verified significant GE intecation effects without main effect for trait Emph. At gene *GSTM*1, TOW-GE, VW-TOW-GE

**Table 3. Summary results of association analysis for *HMOX*1 based on the COPD dataset.** The p-values are shown for testing the gene's main effect (top panel), gene-by-smoking interaction with main effect (middle panel), gene-by-smoking interaction without main effect (bottom panel).

| Gene's main effect | | | | | |
|---|---|---|---|---|---|
| trait | TOW | SKAT | WSS | CMC | VW-TOW |
| GasTrap | 0.3917 | 0.542 | 0.2618 | 0.9038 | 0.5539 |
| ExacerFreq | 0.7050 | 0.6149 | 0.1922 | 0.9845 | 0.8155 |
| Emph | **0.0204** | **0.0328** | **0.0036** | 0.9155 | **0.0360** |
| Pi10 | **0.0283** | **0.0266** | **0.0457** | 0.4501 | 0.0559 |
| EmphDist | 0.8083 | 0.7363 | 0.5172 | 0.7520 | 0.8888 |
| 6MWD | 0.8299 | 0.8451 | 0.8526 | 0.9985 | 0.8642 |
| FEV1 | 0.6825 | 0.6928 | 0.7057 | 0.6906 | 0.7691 |
| COPD | 0.8637 | 0.8277 | 0.8345 | 0.7540 | 0.8677 |
| Gene-by-smoking interaction with main effect | | | | | |
| trait | TOW-GE | ISKAT | WSS | CMC | VW-TOW-GE |
| GasTrap | 0.7432 | 0.8001 | 0.2610 | 0.8894 | 0.8033 |
| ExacerFreq | 0.5883 | 0.2389 | 0.2768 | 0.9964 | 0.3921 |
| Emph | 0.4024 | 0.2718 | 0.1140 | 0.9861 | 0.5696 |
| Pi10 | 0.1208 | 0.4084 | 0.0821 | 0.9948 | 0.0651 |
| EmphDist | 0.5315 | 0.4794 | 0.6006 | 0.9892 | 0.4886 |
| 6MWD | 0.6174 | 0.3624 | 0.4211 | 0.9929 | 0.6793 |
| FEV1 | 0.8656 | 0.7748 | 0.4419 | 0.9575 | 0.9178 |
| COPD | 0.2302 | 0.3029 | 0.9089 | 0.9424 | 0.3394 |
| Gene-by-smoking interaction without main effect | | | | | |
| trait | TOW-GE | ISKAT | WSS | CMC | VW-TOW-GE |
| GasTrap | 0.3388 | 0.5724 | 0.1207 | 0.6040 | 0.4967 |
| ExacerFreq | 0.3818 | 0.2810 | 0.0915 | 0.9320 | 0.4513 |
| Emph | **0.0189** | **0.0487** | **0.0011** | 0.8288 | **0.0349** |
| Pi10 | **0.0304** | 0.0532 | **0.0118** | 0.5587 | 0.0571 |
| EmphDist | 0.8166 | 0.8062 | 0.7610 | 0.8066 | 0.8217 |
| 6MWD | 0.7253 | 0.3463 | 0.6810 | 0.9929 | 0.6811 |
| FEV1 | 0.5604 | 0.7387 | 0.4519 | 0.3043 | 0.7280 |
| COPD | 0.8869 | 0.8877 | 0.8657 | 0.3204 | 0.9300 |

Note: The bold numbers represent p-values of significant tests (significance level = 0.05).

and ISKAT verified GE interaction effect without main effect for trait EmphDist, while all other methods failed in the verification tests. At gene *TGFB*1, TOW-GE, VW-TOW-GE and ISKAT verified GE interaction effect without main effect for trait ExacerFreq (S1 Table). Gene *TNF* was only identified by the modified CMC method and the modified WSS method for gene-by-smoking interaction with main effect (S2 Table). Gene *EPHX*1 was only identified by the modified WSS method for gene-by-smoking interaction with main effect (S4 Table). Four genes with gene-by-smoking interaction effects (*GSTM*1, *HMOX*1, *SFTPB*, and *TGFB*1) were identified by our methods (S1 and S3 Tables, Tables 3 and 4).

## Discussion

Recent evidence shows that gene-environment interactions of rare variants may play an important role in explaining the etiology of a complex disease. However, there are limited methods that can be employed to test the effects of GE interactions for rare variants. In this study, we propose two new methods for testing GE interactions for rare variants only or for

**Table 4. Summary results of association analysis for GSTM1 based on the COPD dataset.** The p-values are shown for testing the gene's main effect (top panel), gene-by-smoking interaction with main effect (middle panel), gene-by-smoking interaction without main effect (bottom panel).

| Gene's main effect | | | | |
|---|---|---|---|---|
| trait | TOW | SKAT | WSS | CMC | VW-TOW |
| GasTrap | 0.2309 | 0.6152 | 0.6163 | 0.2479 | 0.2848 |
| ExacerFreq | 0.7198 | 0.7823 | 0.7677 | 0.9138 | 0.6594 |
| Emph | 0.1401 | 0.3901 | 0.8496 | 0.1686 | 0.2355 |
| Pi10 | **0.0177** | 0.1069 | 0.1705 | 0.0749 | **0.0151** |
| EmphDist | **0.0077** | **0.0256** | 0.3545 | **0.0487** | **0.0082** |
| 6MWD | 0.6011 | 0.6856 | 0.8401 | 0.9260 | 0.5707 |
| FEV1 | 0.1190 | 0.5920 | 0.9013 | 0.2301 | 0.0866 |
| COPD | 0.2144 | 0.2178 | 0.4699 | 0.2078 | 0.3243 |
| Gene-by-smoking interaction with main effect | | | | |
| trait | TOW-GE | ISKAT | WSS | CMC | VW-TOW-GE |
| GasTrap | 0.8652 | 0.7482 | 0.5358 | 0.1096 | 0.9158 |
| ExacerFreq | 0.7417 | 0.0867 | 0.3860 | 0.0599 | 0.5606 |
| Emph | 0.6829 | 0.9901 | 0.6833 | 0.2927 | 0.7207 |
| Pi10 | 0.2757 | 0.5465 | 0.4808 | 0.1164 | 0.4506 |
| EmphDist | 0.1287 | 0.2314 | 0.6639 | 0.6781 | 0.1126 |
| 6MWD | 0.8144 | 0.8769 | 0.8893 | 0.3781 | 0.8384 |
| FEV1 | 0.9389 | 0.4145 | 0.6640 | 0.1277 | 0.9169 |
| COPD | 0.9944 | 0.8870 | 0.7842 | 0.2098 | 0.9878 |
| Gene-by-smoking interaction without main effect | | | | |
| trait | TOW-GE | ISKAT | WSS | CMC | VW-TOW-GE |
| GasTrap | 0.5160 | 0.2723 | 0.8413 | 0.6725 | 0.5769 |
| ExacerFreq | 0.5887 | 0.7348 | 0.6194 | 0.2701 | 0.6796 |
| Emph | 0.1041 | 0.1114 | 0.6514 | 0.4787 | 0.1691 |
| Pi10 | 0.0697 | 0.1112 | 0.1282 | 0.0844 | 0.0631 |
| EmphDist | **0.0071** | **0.0229** | 0.6162 | 0.1078 | **0.0131** |
| 6MWD | 0.7759 | 0.9342 | 0.9903 | 0.8683 | 0.7867 |
| FEV1 | 0.2833 | 0.4709 | 0.6934 | 0.2673 | 0.2254 |
| COPD | 0.3641 | 0.1615 | 0.4693 | 0.5593 | 0.4934 |

The bold numbers represent p-values of significant tests (significance level = 0.05).

both rare and common variants. We employ a generalized linear model to model the relationship between the trait and the GE interactions. Our model focuses on GE interactions by first adjusting for genetic main effects, environmental main effects, and possible covariates. Two methods are designed for different scenarios through specific weigh-selection mechanisms. TOW-GE assigns the majority of weights on rare variants by environment interactions. VW-TOW-GE balances common and rare variants by performing weight assignments separately for common variants by environment interactions and rare variants by environment interactions. Both methods achieve the best possible power with an adaptive weight selection procedure.

In the application, we have tested genetic association for 7 traits of COPD. Our proposed methods verified the most significant GE interactions, especially for gene-by-smoking interactions without main effect and performs the best compared to other methods. In simulation studies, we also demonstrated that our proposed methods perform better in different scenario: with main effect and without main effect. Our results show that the proposed methods

TOW-GE or VW-TOW-GE demonstrate better power in most cases compared with competing methods.

The power of a test varies according to the number of GE interactions of rare or common variants, the effect directions of GE interactions, and the MAFs of variants. When substantial of GE interactions have opposite directions of effects, the quadratic statistics TOW-GE, VW-TOW-GE, and ISKAT are powerful. When effects of GE interactions of common variants play a primary role, CMC is more powerful than ISKAT, WSS, and has similar power to VW-TOW-GE.

In our proposed method, the optimal weights of TOW-GE are derived analytically; thus the computation cost is relatively small. On the other hand, TOW-GE is flexible and allows for prior biological information to be incorporated by using flexible weights, such as weights derived from the expression quantitative trait locus (eQTL), which may further improve the power of TOW-GE. In addition, TOW-GE allows for adjustment of covariates. The covariates could be demographic variables, environmental variables, clinical variables, and/or principle components of genotype scores. The adjustment of covariates makes TOW-GE not only able to eliminate the effect of confounders but also able to correct for possible population stratification in admixed populations. One possible advantage of TOW-GE compared to ISKAT is that TOW-GE utilizes the residuals of both the trait value and the GE interactions, which are obtained by adjusting for covariates from linear regression models, respectively, while ISKAT utilizes only the residual of the trait value.

The proposed test statistic TOW-GE does not have an asymptotic distribution and a permutation procedure is needed to estimate its p-value, which is time consuming compared to methods with asymptotic distributions. To save time when applying the proposed methods in genetic association studies, we can use the "step-up" procedure [20, 21] to determine the number of permutations. This can show evidence of association based on a small number of permutations first (e.g.1,000) and then a large number of permutations are used to test the selected potentially significant genes. Specifically, the computation time of p-value estimation of TOW-GE and VW-TOW-GE for a gene in the real data analysis was about 30 seconds using our R program on 6 Dell PowerEdge C6320 servers. Each server has two 2.4GHz Intel Xeon E5-2680 v4 fourteen-core processors and 600 MB average memory. We have uploaded the R program onto GitHub at https://github.com/Jianjun-CN/Single-GE.

## Supporting information

**S1 Table.**
(PDF)

**S2 Table.**
(PDF)

**S3 Table.**
(PDF)

**S4 Table.**
(PDF)

**S1 File.**
(TEX)

## Acknowledgments

A superior high-performance computing infrastructure at the University of North was used in obtaining results presented in this publication.

## Author Contributions

**Formal analysis:** Zihan Zhao, Jianjun Zhang.

**Methodology:** Zihan Zhao, Han Hao.

**Visualization:** Zihan Zhao, Qiuying Sha.

**Writing – original draft:** Zihan Zhao, Han Hao.

**Writing – review & editing:** Zihan Zhao, Jianjun Zhang, Qiuying Sha, Han Hao.

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
