## [Decision Letter · Decision Letter 0]

4 Dec 2019

PONE-D-19-27761

Testing gene-environment interactions for rare and/or common variants in sequencing association studies

PLOS ONE

Dear Dr. Hao,

Thank you for submitting your manuscript to PLOS ONE. After careful consideration, we feel that it has merit but does not fully meet PLOS ONE’s publication criteria as it currently stands. Therefore, we invite you to submit a revised version of the manuscript that addresses the points raised during the review process.

We would appreciate receiving your revised manuscript by Jan 18 2020 11:59PM. To enhance the reproducibility of your results, we recommend that if applicable you deposit your laboratory protocols in protocols.io, where a protocol can be assigned its own identifier (DOI) such that it can be cited independently in the future. For instructions see: http://journals.plos.org/plosone/s/submission-guidelines#loc-laboratory-protocols

We look forward to receiving your revised manuscript.

Kind regards,

Heming Wang, PhD

Academic Editor

PLOS ONE

Journal Requirements:

The Genetic Analysis workshops are supported by NIH grant R01 GM031575 from the 262

National Institute of General Medical Sciences. Preparation of the Genetic Analysis 263

Workshop 17 Simulated Exome Data Set was supported in part by NIH R01 MH059490 264

and used sequencing data from the 1000 Genomes Project (www.1000genomes.org). 265

This research used data generated by the COPDGene study (phs000179/HMB and 266

phs000179/DS-CS-RD), which was supported by National Institutes of Health (NIH) 267

grants U01HL089856 and U01HL089897. The content is solely the responsibility of the 268

authors and does not necessarily represent the ocial views of the National Heart, 269

Lung, and Blood Institute or the National Institutes of Health. The COPDGene project 270

is also supported by the COPD Foundation through contributions made by an Industry 271

Advisory Board comprised of P zer, AstraZeneca, Boehringer Ingelheim, Novartis, and 272

Sunovion.

Q Sha was supported by the National Human Genome Research Institute (https://www.genome.gov/) of the National Institutes of Health under Award Number R15HG008209. The content is solely the responsibility of the authors and does not necessarily represent the official views of the National Institutes of Health. The funders had no role in study design, data collection and analysis, decision to publish, or preparation of the manuscript.

3. We note you have included a table to which you do not refer in the text of your manuscript. Please ensure that you refer to Table 4 in your text; if accepted, production will need this reference to link the reader to the Table.

Reviewers' comments:

Reviewer's Responses to Questions

**Comments to the Author**

1. Is the manuscript technically sound, and do the data support the conclusions?

Reviewer #1: Yes

Reviewer #2: Yes

2. Has the statistical analysis been performed appropriately and rigorously? 

Reviewer #1: Yes

Reviewer #2: Yes

3. Have the authors made all data underlying the findings in their manuscript fully available?

Reviewer #1: Yes

Reviewer #2: Yes

4. Is the manuscript presented in an intelligible fashion and written in standard English?

Reviewer #1: Yes

Reviewer #2: Yes

5. Review Comments to the Author

Reviewer #1: In this manuscript, authors proposed methods to detect gene-environment (GE) interactions for rare and/or common variants. A generalized linear model was developed and a score test was established for testing GE interactions. Permutation tests were used for obtaining p-values. Simulation studies were conducted to validate the properties of the test. It was showed that the proposed test was more powerful than existing ones in this area. The application of proposed model has been demonstrated with an empirical analysis of data for Chronic obstructive pulmonary disease. It is a nice work and provides biomedical researchers with useful methodology to address scientific questions related to GE interactions, particularly of rare variants.

My questions are about the computation.

1. I understand permutation tests have high computational cost when sample size is large and/or the number of features/genes is large. It was mentioned that HPC infrastructure had been utilized for this study. I wonder if it is feasible to perform the proposed test on a personal computer as it is more convenient for average users.

2. Are the code of your model/test available to public?

Reviewer #2: In this paper, Zhao and colleagues describe a novel method for rare and common variants gene-environment interaction testing in sequencing association studies. The method builds upon substantial existing work and is more powerful than existing methods by using an adaptive weight selection procedure. The text overall is very clearly written.

However, I do have some minor concerns that I would like the authors to address.

(1) The permutation procedure for the p-value calculation is time-consuming. Barnett et.al (JASA, 2017) provided a scheme for permutation test to save computation time. It would be good to incorporate it in the paper.

(2) In the real data application, the authors only test for association of two known genes. It would be great for the authors to show the results of the gene-based analysis of the whole genome.

Reference

[1] Barnett, I., Mukherjee, R. and Lin, X., 2017. The generalized higher criticism for testing SNP-set effects in genetic association studies. Journal of the American Statistical Association, 112(517), pp.64-76.

6. PLOS authors have the option to publish the peer review history of their article (what does this mean?). If published, this will include your full peer review and any attached files.

Reviewer #1: No

Reviewer #2: No

---

## [Author Response · Author response to Decision Letter 0]

7 Jan 2020

Dear Dr. Wang,

We thank you for your encouraging evaluation for our manuscript “Testing gene-environment interactions for rare and/or common variants in sequencing association studies” (Manuscript Number: PONE-D-19-27761R1). We appreciate the editor's patient work in organizing the reviewing process. After thoroughly and carefully addressing the reviewers' comments and making corresponding revisions in the manuscript highlighted in red, we are now resubmitting the paper for your further consideration. Please find our point-by-point response to the reviewers' comments in the letter named Response to Reviewers.

Thank you for the helpful and constructive review. We feel that the manuscript is much improved and hope you now find it suitable for publication in PLOS ONE.

Han Hao

---

## [Decision Letter · Decision Letter 1]

3 Feb 2020

Testing gene-environment interactions for rare and/or common variants in sequencing association studies

PONE-D-19-27761R1

Dear Dr. Hao,

We are pleased to inform you that your manuscript has been judged scientifically suitable for publication and will be formally accepted for publication once it complies with all outstanding technical requirements.

With kind regards,

Heming Wang, PhD

Academic Editor

PLOS ONE

Additional Editor Comments (optional):

Reviewers' comments:

Reviewer's Responses to Questions

**Comments to the Author**

1. If the authors have adequately addressed your comments raised in a previous round of review and you feel that this manuscript is now acceptable for publication, you may indicate that here to bypass the “Comments to the Author” section, enter your conflict of interest statement in the “Confidential to Editor” section, and submit your "Accept" recommendation.

Reviewer #1: All comments have been addressed

Reviewer #2: All comments have been addressed

2. Is the manuscript technically sound, and do the data support the conclusions?

Reviewer #1: Yes

Reviewer #2: Yes

3. Has the statistical analysis been performed appropriately and rigorously? 

Reviewer #1: Yes

Reviewer #2: Yes

4. Have the authors made all data underlying the findings in their manuscript fully available?

Reviewer #1: Yes

Reviewer #2: Yes

5. Is the manuscript presented in an intelligible fashion and written in standard English?

Reviewer #1: Yes

Reviewer #2: Yes

6. Review Comments to the Author

Reviewer #1: (No Response)

Reviewer #2: (No Response)

7. PLOS authors have the option to publish the peer review history of their article (what does this mean?). If published, this will include your full peer review and any attached files.

Reviewer #1: No

Reviewer #2: No

---

## [Editor Report · Acceptance letter]

20 Feb 2020

PONE-D-19-27761R1 

Testing gene-environment interactions for rare and/or common variants in sequencing association studies 

Dear Dr. Hao:

I am pleased to inform you that your manuscript has been deemed suitable for publication in PLOS ONE. Congratulations! Your manuscript is now with our production department. 

With kind regards,

on behalf of

Dr. Heming Wang 

Academic Editor

PLOS ONE